# N6-Isopentenyladenosine Impairs Mitochondrial Metabolism through Inhibition of EGFR Translocation on Mitochondria in Glioblastoma Cells

**DOI:** 10.3390/cancers14246044

**Published:** 2022-12-08

**Authors:** Cristina Pagano, Laura Coppola, Giovanna Navarra, Giorgio Avilia, Sara Bruzzaniti, Erica Piemonte, Mario Galgani, Rosa Della Monica, Lorenzo Chiariotti, Mariella Cuomo, Michela Buonaiuto, Giovanni Torelli, Pasquale Caiazzo, Chiara Laezza, Maurizio Bifulco

**Affiliations:** 1Department of Molecular Medicine and Medical Biotechnology, University of Naples “Federico II”, 80131 Naples, Italy; 2Department of Biology, University of Naples “Federico II”, 80126 Naples, Italy; 3Institute of Endocrinology and Experimental Oncology (IEOS), National Research Council (CNR), 80125 Naples, Italy; 4CEINGE—Biotecnologie Avanzate, Via Gaetano Salvatore 486, 80131 Naples, Italy; 5Neurosurgery Unit A.O. San Giovanni di Dio e Ruggi d’ Aragona, Salerno’s School of Medicine Largo Città di Ippocrate, 84131 Salerno, Italy; 6Osservatorio Oncologico, 84091 Battipaglia, Italy; 7Neurosurgery, Unit A.O. “A.Cardarelli”, 80145 Naples, Italy

**Keywords:** glioblastoma, iPA, mitochondrial metabolism, EGFR, PUMA

## Abstract

**Simple Summary:**

Glioblastomas are aggressive and incurable brain tumors, being resistant to therapy. N6-isopentenyladenosine (i6A or iPA) is a naturally derived molecule that has been studied for its anti-glioma effects. We found that iPA treatment induces an alteration of cellular metabolism due to inhibition of EGFR translocation on mitochondria and activation of cell death following PUMA upregulation. Our findings suggest that inducing dysfunctional mitochondria through iPA might be a promising therapeutic avenue in the treatment of glioblastoma.

**Abstract:**

Glioblastoma multiforme (GBM) is the most aggressive malignant brain tumor and is poorly susceptible to cytotoxic therapies. Amplification of the epidermal growth factor receptor (EGFR) and deletion of exons 2 to 7, which generates EGFR variant III (vIII), are the most common molecular alterations of GBMs that contribute to the aggressiveness of the disease. Recently, it has been shown that EGFR/EGFRvIII-targeted inhibitors enhance mitochondrial translocation by causing mitochondrial accumulation of these receptors, promoting the tumor drug resistance; moreover, they negatively modulate intrinsic mitochondria-mediated apoptosis by sequestering PUMA, leading to impaired apoptotic response in GBM cells. N6-isopentenyladenosine (i6A or iPA), a cytokinin consisting of an adenosine linked to an isopentenyl group deriving from the mevalonate pathway, has antiproliferative effects on numerous tumor cells, including GBM cells, by inducing cell death in vitro and in vivo. Here, we observed that iPA inhibits the mitochondrial respiration in GBM cells by preventing the translocation of EGFR/EGFRvIII to the mitochondria and allowing PUMA to interact with them by promoting changes in mitochondrial activity, thus playing a critical role in cell death. Our findings clearly demonstrate that iPA interferes with mitochondrial bioenergetic capacity, providing a rationale for an effective strategy for treating GBM.

## 1. Introduction

Glioblastoma multiforme (GBM) is the most aggressive primary malignancy of the brain and has a median survival of 12–15 months and a 5-year survival rate of <5% [1]. The standard therapeutic treatment consists in surgical resection of the tumor, followed by radiotherapy and chemotherapy (Stupp protocol), but the high frequency of molecular alterations and the inter- and intra-tumoral heterogeneity renders GBM highly resistant to therapies [2,3,4,5]. Amplification and deletion mutants of the epidermal growth factor receptor (EGFR) gene are frequently found in glioma cancer cells; in particular, the truncation of the EGFR protein extracellular domain, EGFR variant III (vIII), generated by truncation of exons 2 to 7, was only found in GBMs and occurs at an overall frequency of 25–64% [6,7]. These alterations promote proliferation, angiogenesis, invasion, and resistance to therapies [8,9]. Substantial evidence highlights the role of EGFR in intracellular trafficking to sub-cellular organelles, such as mitochondria. EGFR, upon EGF stimulation, can translocate on mitochondria (MitoEGFR) through endocytosis, regulating mitochondria dynamics and enhancing energy production. This event causes the increase of cell motility in vitro and metastasis in vivo in NSLC [10]. Further, it has been observed that EGFR on mitochondria interacts with cytochrome c oxidase subunit II (COXII), which regulates the survival pathway, contributing to oncogenesis [11]. MitoEGFR can modulate several cellular events, such as cellular ATP production, cell drug resistance, and apoptosis, and it might have a regulatory role in autophagy [12]. In addition, EGFRvIII can translocate to mitochondria dependently by Src activation, promoting tumorigenicity. One study reported that it decreases glucose dependency and increases the oxidative phosphorylation (OXPHOS), thereby favoring resistance to targeted therapy [13]. Moreover, recent research has revealed that EGFR-targeted inhibitors in glioblastoma cell lines increase mitochondrial translocation of both EGFR and EGFRvIII, causing mitochondrial accumulation of these receptors and contributing to tumor drug resistance, thus providing evidence for a connection between the mitochondrial EGFR pathway and apoptosis. The same authors found that EGFR and EGFRvIII interact with pro-apoptotic protein PUMA, preventing its translocation on mitochondria and increasing aggressiveness of the tumor [14,15]. Indeed, one study reported that the pro-apoptotic protein PUMA mediates the apoptosis induced by EGFR inhibitors in head and neck cancer cells [16], while another reported that the translocation of PUMA to the mitochondria can provoke cell death in lymphoma cells [17].

N6-isopentenyladenosine (i6A or iPA) is a natural cytokinin consisting of an adenosine linked to an isopentenyl group deriving from the mevalonate pathway [18]. This compound has antiproliferative effects in numerous cancer cells by inducing apoptosis in vitro and in vivo, hindering the angiogenic process, migration, and vasculogenic mimicry, including in GBM cell lines and primary GBM patient-derived cells [19,20,21,22]. In particular, in U87MG cells, we described the ability of iPA to arrest cell viability through inhibition of the EGFR signaling pathway [23]. In this study, to represent the molecular heterogeneity characterizing this tumor, we used U87MG, and the same ones engineered to over-express wild type (wt) EGFR or EGFRvIII and primary GBM patients-derived cells. We showed for the first time that iPA caused mitochondrial dysfunction, preventing the translocation of EGFR/EGFRvIII on the mitochondria by inhibition of tyrosine residue Y845 phosphorylation in both receptors. This event allows PUMA to interact with the organelle, thus promoting alterations in the mitochondrial dynamic and morphology, causing cell death in glioblastoma cells.

## 2. Materials and Methods

### 2.1. Cell Cultures and Reagents

Glioblastoma multiforme cell lines U87MG were purchased from Elabscience (Elabscience, Houston, TX, USA; catalog No. EP-CL-0238); U251MG cells were purchased from Elabscience (Elabscience, Houston, TX, USA; catalog No. EP-CL-0237); normal human cells derived from human brain tissue, NHAs, were purchased from Lonza (Product Code: CC-2565); U87MG expressing EGFRwt and U87MG expressing EGFRvIII were kindly donated by Professor F.B. Furnari of the Ludwig Institute for Cancer Research and the Moores Cancer Center, University of California, San Diego, CA, USA [24]. Cells were grown in DMEM (Gibco, Thermo Fisher Scientific, Monza, Italy) supplemented with 10% heat inactivated fetal bovine serum, 1% non-essential amino acids (Lonza, Rome, Italy), 1% sodium pyruvate, 1% L-glutamine, and 0.1% Plasmocin^TM^ (InvivoGen, San Diego, CA, USA). All cell cultures were cultured at 37 °C in humidified 5% CO_2_ controlled atmosphere. For this paper, cells were treated with N6-isopentenyladenosine (iPA) (Cayman Chemical Company, Cod: 20522).

### 2.2. Preparation of Glioblastoma Primary Cell Lines

Primary cell lines were isolated from surgically removed tumor sections obtained from patients recovered at the Neurosurgery Service of “Antonio Cardarelli” Medical Hospital (Naples, Italy). A second sample was taken from each patient and used for clinical diagnosis, according to the International Classification of CNS tumors drafted under the auspices of the World Health Organization (WHO). All tissue samples were collected in accordance with the ethical standards of the Institutional Committee (DEL. No. 897, 13 August 2020). Informed consent in written form was obtained from all subjects involved in the study. Briefly, tumor specimens (designated as GBMn) were minced into small fragments by surgical scalpels. Subsequently, further tissue disaggregation of samples was obtained with the gentleMACS™ Dissociator in combination with the Tumor Dissociation Kit, human (Miltenyi Biotec, Cologne, Germany. Cod#130-095-929), using the appropriate enzymes mix, according to the protocol. The obtained cell suspension was then forced through a MACS SmartStrainer, mesh size 70 μm, placed on a 50 mL tube. The cell suspension was centrifuged at 300× *g* for 5 min. The supernatant was completely aspirated, and the cells cultured in DMEM/Ham’s F-12 supplemented with 15% heat-inactivated fetal bovine serum, 2% L-glutamine, 1% sodium pyruvate, 1% non-essential amino acids (Lonza, Rome, Italy), and 1.5% D-glucose.

### 2.3. Western Blot Analysis

Cells were grown in p60 plates at a density of 2 × 10^5^ cells/cm², treated with iPA diluted in growth medium, and subsequently harvested. Cells were washed twice in phosphate-buffered saline (PBS) and lysed in cold RIPA lysis buffer (50 mM Tris-HCl, 150 mM NaCl, 0.5% Triton X-100, 10 mg/mL leupeptin, 0.5% deoxycholic acid, 2 mM phenylmethylsulfonyl fluoride, and 10 mg/mL aprotinin containing protease and phosphatase inhibitors) (Sigma-Aldrich, St. Louis, MO, USA). Protein concentration was determined using Protein Analysis Dye Reagent Concentrate (BioRad, Hercules, CA, USA). After resuspension in Laemmli sample buffer, proteins were loaded and separated on SDS-PAGE gels at different percentage, transferred to nitrocellulose membranes (Trans-Blot Turbo Transfer Pack, BioRad, Hercules, CA, USA) using a Trans-Blot Turbo (BioRad, Hercules, CA, USA), saturated with 5% fat-free milk in Tris saline buffer containing 0.1% Tween-20 (TBST) for 1h, and probed overnight at 4 °C with primary specific antibodies. After 1 h of incubation with horseradish peroxidase-conjugated goat anti-mouse or anti-rabbit IgG (BioRad, Hercules, CA, USA), signals were detected using the BioRad Chemidoc MP image sensor after the membranes were soaked in enhanced ECL reagents (Amersham, GE Healthcare, Buckinghamshire, UK). Some membrane signals were captured by exposure to X-ray film (Santa Cruz Biotechnology, CA, USA). Primary antibodies used for the western blot analysis: mouse monoclonal anti-human p-EGFR(Tyr 845) (sc-57542), mouse monoclonal anti-human Mfn1 (sc-166644), mouse monoclonal anti-human Fis1 (sc-376447), mouse monoclonal anti-human β-actin (sc-47778), and rabbit polyclonal anti-human HA (sc-805) were purchased from Santa Cruz Biotechnology (Santa Cruz, United States); mouse monoclonal anti-human SDHA (459200) was purchased from Thermo Fisher; rabbit polyclonal anti-human Puma (4976) and rabbit monoclonal anti-human EGF receptor (2646S) were purchased from Cell Signaling Technology.

### 2.4. Immunofluorescence

Glioblastoma cells were grown on glass cover slips in 24-well plates at a density of 1 × 10^5^ cells/cm² and allowed to adhere for 24 h. Cell medium was then replaced, and cells were treated with iPA 10 µM for 24 h. Afterwards, cells were fixed with 4% paraformaldehyde, permeabilized with 0.2% Triton X-100, and blocked using PBS-BSA 0.4%. Cells were then incubated with anti-TOM20, anti-EGFR, and anti-PUMA at 4 °C overnight. Following washes with PBS, cells were incubated with a labeled secondary antibody at room temperature for 1 h. Nuclei were then stained with DAPI (Hoechst 33258, Ref H3569, Invitrogen by Thermo Fisher Scientific). Finally, cells were again washed with PBS and mounted on the slide using Dako Fluorescent Mounting Medium. The images were acquired using a Leica DMi8 M System fluorescence microscope (Leica Microsystems, Milan, Italy). The following antibodies were used for immunofluorescence analysis: mouse monoclonal anti-human Tom20 (sc17764) (Santa Cruz Biotechnology, Santa Cruz, Dallas, TX, USA); rabbit monoclonal anti-human EGF Receptor (2646S) (Cell Signaling Technology, Danvers, MA, USA); goat monoclonal anti-human PUMA (NBP1-52093) (Novus Biologicals); secondary antibody: Alexa Fluor^®^ 488 (711-545-152) goat polyclonal anti-rabbit IgG (Jackson ImmunoResearch, Cambridge, UK), DyLight^®^ 594 (ab96873) goat polyclonal anti-mouse IgG (Abcam, Cambridge, UK), DyLight^®^ 350 (A21081) donkey polyclonal anti-goat (Invitrogen, Carlsbad, CA, USA). MitoTrackerTM Green FM (M7514) (Invitrogen, Carlsbad, CA, USA) was used for mitochondria visualization.

### 2.5. Mitochondrial Protein Fractionation

Mitochondrial organelles were isolated from 2 × 10^7^ cells treated with iPA 10 µM for 24 h using the Mitochondrial Isolation Kit for Cultured Cells (Thermo Fisher Scientific, Monza, Italy, COD# 89874). Mitochondrial preparations were obtained following the manufacturer’s instructions and resolved by SDS-PAGE electrophoresis.

### 2.6. PUMA Depletion by RNA Interference

PUMA and control-siRNA (IDT DsiRNAs TriFECTa^®^ Kits) were used for transfection of U87MG cell line. Cells were seeded in 96 plates at a density of 1 × 10^4^ per well for Seahorse Analysis or in p60 dishes at a density of 1.5 × 10^6^ cells/cm^2^ for protein extraction. After 24 h, both PUMA siRNA and scramble siRNA were delivered into the cell cultures via Oligofectamine reagent (Invitrogen, CA, USA, #12252-011), according to the manufacturers’ instructions. The cells were incubated with the transfection reagents for 4 h and treated with iPA 10 µM for 18 and 24 h. The cells were then harvested for subsequent analysis of protein knockdown. Experiments were performed three or more times to ensure reproducibility and statistical fidelity.

### 2.7. Primary Glioblastoma Characterization

To characterize primary glioblastoma, DNA methylation status of each tumor sample was evaluated in 850,000 CpG sites using the EPIC ARRAY 850 Beads-Chip (850 K), according to the manufacturer’s instructions. The epigenomic profile was compared to a reference cohort previously analyzed at the German Cancer Research Center using a specific algorithm and customized bioinformatics packages as described previously [25]. A copy number variation profile is calculated starting from the array data [26]. In this study, sample copy number variation profiles were used to verify the presence/absence of the EGFR gene. “Gain” or “amplification” was determined by log2 > 0.3. The molecular characteristics of the primary cell lines here used are reported in Table 1.

### 2.8. IDH1 and IDH2 Mutation Status

Tumor DNA was isolated from a formalin-fixed, paraffin-embedded (FFPE) sample using the QIAamp DNA FFPE Tissue Kit (Qiagen s.r.l.). DNA was then amplified via PCR using specific primers designated for exon 4 of IDH1 and IDH2 genes.

IDH1:

Forward primer: 5′-TGTAAAACGACGGCCAGTGGATGCTGCAGAAGCTATAA-3′;

Reverse primer: 5′-CAGGAAACAGCTATGACCTTCATACCTTGCTTAATGGG-3′.

IDH2:

Forward primer: 5′-TGTAAAACGACGGCCAGTAATTTTAGGACCCCCGTCTG-3′;

Reverse primer: 5′-CAGGAAACAGCTATGACCGGGGTGAAGACCATTTTGAA-3′.

Sanger sequencing method was used for analysis of codon 100 and 132 of IDH1 and codon 140 and 172 of IDH2 [22].

### 2.9. MGMT Methylation Assessment

Methylation-specific PCR (MSP) was used for the determination of MGMT promoter methylation status. DNA extracted from tumor specimens was converted using sodium bisulfite, following the EZ DNA Methylation Gold Kit protocol (Zymo Research). Methylation-specific PCR was performed using nested PCR. The first PCR was performed using specific primers:

Forward primer: 5′-GGATATGTTGGGATATAGTT-3′,

Reverse primer: 5′-CCATCCACAATCACTACAA-3′.

The second PCR step made use of different primers for methylated and non-methylated DNA samples:
-Methylated MGMT:
forward primer 5′-TTTCGACGTTCGTAGGTTTTCGC-3′,reverse primer 5′-GCACTCTTCCGAAAACGAAACG-3′.-Unmethylated MGMT:
forward primer 5′-TTTGTGTTTTGATGTTTGTAGGTTTTTGT-3′,reverse primer 5′-AACTCCACACTCTT CCAAAAACAAAACA-3′.

These primers were chosen as previously described [27]. The positive control used was a commercial methylated DNA, specific for methylated MGMT alleles. Similarly, a non-methylated commercial control was used as a negative control (EpiTEC controls from Qiagen S.r.l., Milano, Italy). In each set of the MSP assay, controls without DNA were used. Then, each MSP product was loaded directly onto 3% percent agarose gel, stained with ethidium bromide (Sigma-Aldrich, St. Louis, MO, USA), and examined under ultraviolet illumination using a ChemiDoc MP image sensor (BioRad, Hercules, CA, USA).

### 2.10. Seahorse Analysis

Metabolic profile was evaluated in glioma stabilized cell lines (U87MG, U87MG-EGFRwt, U87MG-EGFRvIII, U251MG), normal human astrocytes (NHA cells), and glioblastoma primary cell line (designed as GBM4), treated or untreated with iPA for 24 h. Real-time measurements of oxygen consumption rate (OCR) were performed by an XFe-96 Analyzer (Agilent Technologies, CA, USA). Specifically, cells were plated in XFe-96 plates (Agilent Technologies, CA, USA) at a concentration of 2 × 10^4^ cells/well and cultured either with DMEM medium or DMEM/Ham’s F-12 medium. OCR was measured in XF DMEM medium (supplemented with 10 mM glucose, 2 mM L-glutamine, and 1 mM sodium pyruvate) under basal conditions and in response to 5 μM oligomycin, 1.5 μM of carbonylcyanide-4-(trifluoromethoxy)-phenylhydrazone (FCCP), and 1 μM of antimycin A and rotenone (all from Sigma-Aldrich, St. Louis, MO, USA). ECAR was measured in XF DMEM medium (Agilent Technologies, CA, USA) under basal conditions and in response to 10 mM glucose, 5 μM oligomycin, and 100 mM of 2DG (all from Sigma-Aldrich, St. Louis, MO, USA). Experiments with the Seahorse were done with the following assay conditions: 3 min mixture; 3 min wait; and 3 min measurement. For each sample, we prepared 3 technical replicates and performed 3 independent experiments. Parameters of the mitochondrial respiration pathway were calculated from OCR profile: basal respiration (calculated as the difference between OCR before the first injection and OCR after Ant + Rot injection), maximal respiration (calculated as the difference between OCR after FCCP injection and OCR after Ant + Rot injection), ATP production (calculated as the difference between OCR before oligomycin injection and OCR after oligomycin injection), and spare capacity (calculated as the difference between maximal respiration and basal respiration).

### 2.11. Cell Transfection

Cell transfections were performed using the FuGene HD reagents (Promega Italia s.r.l., Milano, Italy) following the manufacturer’s protocol. Cells were harvested 24  h after transfection. The plasmid used in this work were the following: EGFR GFP (Plasmid #32751), pHA-PUMA (Plasmid #16588) purchased from AddGene (Watertown, MA, USA) [28,29]; plasmid pLERNL (ΔEGFR) was kindly donated by Professor F.B. Furnari of the Ludwig Institute for Cancer Research and the Moores Cancer Center, University of California, San Diego, La Jolla [30]; plasmid pcDNA-Y845F-EGFR was kindly donated by Professor Julie Boerner of Wayne State University, Detroit, MI [11].

### 2.12. Statistical Analysis

Statistical analyses were carried out by the GraphPad Prism 7.0 software for Windows (GraphPad Software, San Diego, CA, USA) for all the experiments here shown. Obtained data are reported as mean ± S.D. Statistical significance was assessed by two-tailed Student’s *t*-test. For independent groups, we used one-way ANOVA statistical analysis followed by Bonferroni correction for multiple comparisons. *p*-Values * *p* < 0.05, ** *p* < 0.01, and *** *p* < 0.001 were considered significant. All experiments were performed in triplicate and repeated from 3 to 5 times.

## 3. Results

### 3.1. iPA Impairs the Mitochondrial Metabolism in GBM Cell Lines

In our previous study [31], we showed that increasing concentrations of iPA ranging from 0 to 10 μM reduced the proliferation in U87MG and the same ones engineered to over-express wild type (wt) EGFR or EGFRvIII after 48 h of treatment, while no significant effect was observed for the normal human astrocytes cells, NHA cells. Afterwards, we evaluated the degree of cell death induction after iPA treatment of the GBM cell lines. We observed that iPA treatment at 10 μM for 48 h caused necroptosis (Appendix A) in GBM cells lines through the activation of necroptosis markers and induction of PUMA, while in treated NHA cells, apoptotic events were barely detectable, suggesting the specificity of iPA necroptotic effect in tumor cells [31]. Based on our recent study and substantial evidence in the literature about the role of EGFR and PUMA on mitochondria activity, we wondered whether the iPA-induced death of GBM cells was due to an alteration of the mitochondrial metabolism mediated by a modulation of the protein levels of EGFR/EGFRvIII and PUMA or by their localization on the mitochondria. 

First, we evaluated the EGFR and PUMA protein levels in total lysates of GBM cell lines treated with or without iPA for 24 h by western blot analysis. The iPA concentration used for the experiments was 10 μM, for which we observed a poor inhibition of cell proliferation in U87MG cells and over-expressing EGFRwt/EGFRvIII cells at 24 h as determined by BrdU incorporation assay and as described in our previous study [31] (Appendix A). All in vitro experiments were carried out in tri/pentaplicates for iPA treatments, while controls were treated with vehicle 1% DMSO. As depicted in Figure 1A, the overall EGFR levels were higher in cell lines over-expressing EGFR/EGFRvIII than in U87MG; the iPA treatment did not change the EGFR protein levels, while it significantly increased PUMA ones as compared to untreated samples. 

Afterwards, we examined the mitochondrial activity upon iPA treatment using an oxygen consumption rate (OCR) analysis with a Seahorse Extracellular Flux XF24 Analyzer. The analysis showed the overall OCRs of U87MG, U87MG-EGFRwt, and U87MG-EGFRvIII cells were decreased when treated with iPA compared to vehicle-treated cells (Figure 1B–D). To gather further insight into this mechanism, we also evaluated the response of iPA treatment in normal human astrocyte (NHA) primary cells and, in an additional glioblastoma cell line, U251MG, harboring a mutated p 53 (R273H) that lacks the DNA-binding activity and is therefore incapable of activating target genes, including PUMA [32]. First, we evaluated the iPA effect on cell growth of U251MG cells and observed that the iPA treatment of these cells for 48 h at increasing concentration (0 to 10 μM) arrested the cell viability (Appendix A). Annexin V/PI double-staining and flow cytometry analysis revealed that cell death resulted from apoptosis rather than necrosis after 48 h of iPA treatment at 10 μM (Appendix A). Therefore, we investigated the protein levels of EGFR and PUMA in total lysates of NHA and U251MG cells treated with iPA at 10 μM for 24 h through western blot assay. As shown in Figure 2A, in NHA cells, we observed low levels of EGFR [33] in untreated cells and a slight increase of PUMA. In U251MG-treated cells, the results showed low levels of EGFR in both untreated and iPA-treated samples, while PUMA was not detectable (Figure 2A). 

Then, we performed O_2_ consumption rate (OCR) measurements of NHA and U251MG cells after 24 h of treatment with iPA at 10 μM (Figure 2B,C). We did not observed significant changes in the mitochondrial respiration in both treated cell lines as compared to the control groups. Based on these results, we wanted to clarify the role of EGFR and PUMA proteins in mitochondrial activity, performing experiments of over-expression of PUMA (HA-tagged) and EGFR by transient co-transfection with plasmids encoding such proteins in U251MG cells. After 24 h of transfection, we detected high levels of both PUMA, revealed by HA antibody, and EGFR proteins compared with U251MG co-transfected with control plasmid vector (Mock) (Figure 2D). We then analyzed the mitochondrial activity in U251MG that were co-transfected and treated or untreated with iPA at 10 μM for 24 h. As depicted in Figure 2D, iPA-treated U251MG cells showed a reduction in basal and maximal mitochondrial respiratory capacities compared with untreated samples. These results suggest a role for EGFR and PUMA in the regulation of mitochondrial activity.

### 3.2. iPA Inhibits the EGFR/EGFRvIII Translocation on Mitochondria in GBM Cells

A growing body of evidence indicates that EGFR directly regulates mitochondria by translocating to the organelle and modulating several cellular events, cell drug resistance, autophagy, and apoptosis [10,11,12]. On the other hand, PUMA induction is generally associated with mitochondrial localization and apoptosis [14,15]. To determine whether iPA treatment affected the EGFR/EGFRvIII or PUMA translocation to mitochondria, we investigated their localization in the organelle in U87MG and U87MG stably over-expressing EGFR/EGFRvIII cell lines. To this end, biochemical fractionation was undertaken by using variable-speed centrifugation as described in Materials and Methods. Purified mitochondria were resolved by SDS-PAGE with equal loading confirmed by immunoblotting for the mitochondrial marker antibody, SDHA protein. Instead, we used β-Actin as a marker of the cytosolic fractions (Figure 3A). Fractions were analyzed by western blot assay for the presence of EGFR, EGFRvIII, and PUMA using specific antibodies. Figure 3A shows that, in untreated (Ctr) GBM cell lines, high levels of EGFR were found in the mitochondrial fraction, while PUMA was predominantly localized in the cytosolic fraction under the unstressed condition, undergoing only a modest mitochondrial translocation. 

In addition, we observed that the mitochondrial fraction is relatively free of contaminating membranes from lysosomal and endosomal markers. In GBM cells treated with iPA 10 µM for 24 h, we observed an increase of PUMA protein levels in the mitochondrial fraction; in contrast, the iPA treatment completely ablated localization of EGFR on the mitochondria. To ascertain EGFR and PUMA mitochondrial localization, we performed immunofluorescence experiments on U87MG cells by using TOM20 antibody as a mitochondrial marker, EGFR, and PUMA antibodies. As depicted in Figure 3B, in untreated U87MG cells, the EGFR antibody (green signal) was distributed in the membrane and cytoplasm, while the signal of PUMA antibody (blue) was very weak. In merged images, TOM20 signal (red) and EGFR signal (green) co-localized, producing a yellow color. In U87MG treated with iPA 10 μM for 24 h, the PUMA signal (blue) was increased. In merged images, TOM20 signal (red) and PUMA (blue) co-localized, producing a violet color, while the EGFR signal (green) did not overlap with PUMA or TOM20 signals. These data suggest that EGFR could translocate to the mitochondria in the unstressed condition (Ctr), while iPA treatment inhibited EGFR translocation and induced an increase of PUMA protein interacting with the organelle.

### 3.3. iPA Inhibits the Y845 Phosphorylation of EGFR and EGFRvIII in GBM Cells

EGFR undergoes autophosphorylation on several tyrosine residues to be activated, and the phosphorylation of these few tyrosine residues is mediated by several kinases, including Src kinase. This kinase is responsible of Y845 phosphorylation, which is important for the EGFR/EGFRvIII translocation to mitochondria, contributing to their integrity as much as to the survival of cancer cells [34]. Since, in our recent study, we demonstrated that iPA 10 μM at 24 h inhibited the Src Kinase phosphorylation in several GBM cells, including U87MG cells lines [22], we wondered whether iPA treatment might affect the phosphorylation status of Y845 of the EGFR/EGFRvIII and consequentially inhibit their translocation on mitochondria. We performed a 24 h time course of the phosphorylation status of Y845 in U87MG, U87MG-EGFRwt, and U87-EGFRvIII cells treated with iPA 10 μM in comparison with untreated cells and revealed the results through western blot analysis, using a specific antibody recognizing EGFR phosphorylated on tyrosine 845, pEGFRY845. As shown in Figure 4A, we observed that Y845 phosphorylation decreased after 6 h, and at 18 h and 24 h was completely inhibited. We then determined whether the EGFR translocation on mitochondria is dependent on the Y845 phosphorylation. For this purpose, GBM cells were transiently transfected with plasmids encoding a mutant EGFR bearing a phenylalanine (F) substitution at tyrosine (Y) residue 845 (EGFRY845F plasmid for U87MG and U87MG-EGFRwt cells, EGFRvIIIY845F plasmid for U87MG-EGFRvIII cells). Consequently, we purified the mitochondria to analyze the localization of EGFR and PUMA protein in the organelle. 

Figure 4B depicts western blot results from the mitochondrial fractions. High levels of EGFR were revealed in cells transfected with vector control (Ctr), while cells transfected with EGFR/EGFRvIII mutated in Y845 showed no detectable signal, suggesting that Y845 phosphorylation is crucial for EGFRwt/EGFRvIII localization on mitochondria, in line with the published literature [34]. PUMA protein levels were clearly increased in transfected cell lines with the Y845F variant of EGFR/EGFRvIII, similar to the results obtained in the GBM cells treated with iPA. Interestingly, in the cytosolic fractions, the EGFR protein levels were increased in the GBM cells transfected with the Y845F variant of EGFR/EGFRVIII compared to the control cells, but they were not revealed by the specific antibody (pEGFRY845) that recognizes phosphorylated EGFR on tyrosine 845 (Figure 4B). Moreover, PUMA showed only a moderate increase in protein levels in cytosolic fractions of the transfected cells compared to cells transfected with the control vector. To examine the EGFRY845F and PUMA localization on the mitochondria, we performed immunofluorescence experiments in U87MG cells transfected with Y845F variant of EGFR by using the mitochondrial marker TOM20, EGFR, and PUMA antibodies. As depicted in Figure 4C, in U87MG cells transfected with the control vector (Mock), the EGFR signal (green) was localized in the membrane and in cytosol areas. Interestingly, the EGFR signal was localized in the same cytosolic region of TOM20 signal (red). Indeed, the EGFR signal (green) overlapped the TOM20 signal (red), producing a light-yellow color (merge image). In contrast, in U87MG cells transfected with EGFRY845F plasmid, we observed the most widespread EGFR signal (green), and the EGFR signal did not overlap the TOM20 signal (red) (Figure 4C merge). Conversely, the PUMA signal (blue) was increased and overlapped the TOM20 signal (red), producing a violet color in the cytosol. Together, these data suggest that iPA treatment inhibited the phosphorylation of the Y845 site in the EGFR and that this event was crucial in the decision to direct it to the mitochondria.

### 3.4. The Y845 Phosphorylation of EGFR Is Important for Mitochondrial Activity

To assess the EGFR/EGFRvIII Y845 phosphorylation role in the mitochondrial activity, we performed OCR measurement assays in U87MG, U87MG-EGFRwt, and U87MG-EGFRvIII cells transiently transfected with the Y845F variant of the EGFR/EGFRvIII. The analysis by western blot assay revealed that in the transfected GBM cells, the levels of EGFR protein were increased compared to those transfected with the vector control (Ctr and iPA samples), but, interestingly, it was not recognized by the pEGFRY845 antibody. Furthermore, iPA treatment inhibited Y845 phosphorylation of endogenous EGFR in cells transfected with control vector (Mock) (Figure 5A–C, right panels). GBM cells transfected with Y845F variant of EGFR/EGFRvIII as well as iPA-treated ones showed a significantly reduced oxygen consumption rate as compared to control cells (Mock) (Figure 5A–C, left panels), suggesting that the Y845 phosphorylation of EGFR is important for the mitochondrial respiration. To further corroborate the role of Y845 phosphorylation in mitochondrial activity, we co-transfected U251MG cells with EGFRY845F and PUMA plasmids and then performed the OCR analysis. In Figure 5D, the left panel shows that the basal and maximal mitochondrial respiratory capacities and ATP production were significantly decreased in U251MG cells that had been co-transfected with EGFRY845F and PUMA plasmids compared to those transfected with the vector control and treated or untreated with iPA. Western blot results depicted in the right panel of Figure 5D show an over-expression of both EGFR and PUMA (HA-tagged) in co-transfected U251MG cells compared with cells transfected with the vector control. In the same panel, we observed that iPA treatment reduced the Y845 phosphorylation of endogenous EGFR, while in cells co-transfected with PUMA and EGFRY845, the receptor was not revealed by the pEGFRY845 antibody. These findings suggested a critical role of PUMA in mitochondrial activity. 

To ascertain this hypothesis, we evaluated the effects of PUMA knockdown in U87MG cells through small interfering RNA (siRNA). To this end, we transiently transfected U87MG cells with siRNA-PUMA and then treated them with iPA 10 μM for additional 24 h. Subsequently, we performed an oxygen consumption rate (OCR) analysis. As depicted in the right panel of Figure 5E, the western blot of total lysates showed that in U87MG transfected with siRNA-PUMA, the PUMA protein was mostly suppressed after 24 h as compared to the control and iPA-treated cells; also, iPA treatment of transfected cells inhibited the Y845 phosphorylation of EGFR. The analysis of OCR curves revealed that PUMA knockdown inhibited the iPA effect on all parameters of mitochondrial activity (Figure 5E, left panel). Indeed, in the cells transfected with siRNA-PUMA and treated or untreated with iPA, we did not observe significant changes in the basal, ATP production, and maximal respiration as compared to the cells treated with iPA, which totally reduced the mitochondrial activity. These data suggest a protective role of EGFR and a suppressive role of PUMA, as the induction of this protein caused the reduction of oxidative phosphorylation, as described in literature [35]. 

Lastly, because we recently observed that PUMA induction by iPA treatment is an activator of the necroptotic cell death pathway, we wondered whether the PUMA induction could cause changes in mitochondrial morphology [36]. For this reason, we sought to determine whether iPA treatment affected the levels of proteins related to mitochondrial fission, such as Fis1 (fission 1 protein) and fusion as Mfn1 (mitofusin 1) [37]. As shown in Figure 6A, we observed a significant increase in the levels of Fis1 and no change in the protein level of Mfn1 in purified mitochondria of GBM cell lines treated with iPA 10 µM for 24 h as compared to untreated samples. Based on these data, we examined whether iPA affects mitochondrial morphology by immunofluorescence experiments. As shown in Figure 6B,C we observed an obvious morphological change in the mitochondria in U87MG cells stained with MitoTracker (Figure 6B) or by using TOM20 antibody (Figure 6C). The U87MG cells treated with iPA contained more round or fragmented mitochondria, while untreated cells contained more fused or elongated mitochondria.

### 3.5. iPA Selectively Inhibits Mitochondrial Metabolism in GBM Primary Cells

As inter- and intratumor heterogeneity represents an obstacle to broadly efficacious GBM therapy, we also considered whether iPA would affect the viability of a spectrum of clinically relevant GBM cellular subtypes. For this purpose, we established a panel of different primary GBM cell models (named GBM12, GBM21, GBM22, GBM26, and GBM4) from fresh primary tumor resection of glioma patients. We also performed methylome profiling of glioma tissues (GBM WHO IV), and the primary GBM cells derived from the tissues were characterized for the mutational status of IDH1/IDH2, p53, and methylation of MGMT (Table 1) as described in Materials and Methods.

To establish the physiological relevance of the experimental features, we used primary tumor cells at an early passage (up to the third passage) and low oxygen tension (5% normoxia). To gain insight into the ex vivo effects of iPA, we studied the antiproliferative potential of iPA at 10 μM for 24 h and 48 h by performing a BrdU incorporation assay in human primary glioblastoma cells. We observed an inhibition of cell viability of GBM cells after 48 h of treatment (Appendix A). Subsequently, to assess whether the effects observed following iPA treatment were ascribable to apoptotic events, we stained GBM cells with Annexin-V /FITC. Cells were collected after 48 h of treatment, and the results obtained showed that iPA induces apoptosis at 48h of treatment at a concentration of 10 μM (Appendix A).

Then, we explored whether iPA treatment affected the mitochondrial respiration in all GBM primary cells using a Seahorse Extracellular Flux XF24 Analyzer. Similar to the results obtained on GBM cell lines, the O_2_ consumption rate (OCR) measurements revealed a significant decrease of maximal respiration of all primary cells following iPA treatment for 24 h (Figure 7A–D, left panels). Western blot panels (Figure 7A–E, right panel) showed the effects of the treatment on EGFR and PUMA protein levels and on the Y845 phosphorylation. The total lysates of GBM cells treated with iPA showed a reduction in Y845 phosphorylation of EGFR and an increase in PUMA protein levels, while EGFR levels remained unchanged. To corroborate the Y845 phosphorylation of EGFR role in the mitochondrial activity, we transiently transfected GBM4 cells with Y845F variant of EGFR and analyzed the OCR curves of this primary cell line.

Figure 7F left panel shows that basal respiration, ATP production, and maximal respiration and spare capacity were significantly decreased in GBM4 cells transiently transfected with EGFRY845F plasmid compared with the ones transfected with the vector control. Analysis by western blot assay revealed that in the GBM4 cells transfected with EGFRY845F plasmid, the levels of EGFR protein were increased compared to those transfected with the vector control, but, interestingly, this was not recognized by the pEGFRY845 antibody (Figure 7F right panel). These results suggest that the inhibitory effect of iPA on respiration mitochondrial of human primary glioblastoma cells could be due to the similar molecular mechanism observed in GBM cell lines.

Finally, we observed the effects of iPA 10 μM on EGFR and PUMA levels on GBM4 through immunofluorescence staining (Figure 7G). The captured images showed similar results to those obtained on stabilized cell lines; GBM4 treated samples showed an increased blue signal (PUMA), while in merged images, TOM20 co-localized with PUMA, producing a violet color due to the reduction of the EGFR green signal.

## 4. Discussion

Glioblastoma multiforme represents the most lethal and incurable brain tumor, and effective therapy remains a goal for clinical practice. The discovery of new, effective compounds that may be used with success in glioblastoma treatment is an important challenge, particularly due to its chemo-and radio-resistance. In this study, we demonstrated for the first time an interesting anti-glioblastoma activity of the isoprenoid derivative iPA. In our previous research, we observed that this natural cytokinin displays pleiotropic properties able to interfere with tumor growth through several mechanisms of action, including the inhibition of proliferation, the blocking of the cell cycle, and the induction of apoptosis in vitro and in vivo [18,19,20,21,22,23]. iPA can be monophosphorylated into 5′-iPA-monophosphate (iPAMP) by the adenosine kinase (ADK), activating AMP-activated protein kinase (AMPK) in endothelial cells, thus inhibiting the angiogenic process both in vitro and in vivo [19]. Moreover, we observed that it was able to exert anti-inflammatory effects both in vitro and in vivo by directly targeting NK cells, providing a novel pharmacological tool in diseases characterized by a deregulated immune response such as cancer [20]. This drug reduced vasculogenic mimicry process in glioblastoma cells through inhibition the Src/p120-catenin pathway and inhibition of RhoA-GTPase activity [22]. In the glioma cells, the EGFR signaling pathway was decreased via reduction of the STAT3, ERK, and AKT cascade [23], while iPA inhibited the farnesyl diphosphate synthase (FDPS) activity via reduction of protein prenylation, thus arresting the proliferation of thyroid cancer cells [18]. This impaired the autophagic process in melanoma both in vitro and in vivo through AMPK activation and inhibition of the mTOR pathway [38]. Recent and interesting evidence shows that iPA inhibits glioma-initiating cells and induces autophagic cell death, suggesting that i6A is a promising therapeutic molecule to target GICs [39]. We showed that iPA inhibits several signaling pathways of cell proliferation and metabolism, such as RAS, ERK, AKT, and mTOR pathways, which are notoriously mediated by EGFR signaling from the plasma membrane [40]. 

Evidence has also begun to emerge regarding the role of EGFR and EGFRvIII localization in sub-cellular compartments such as as mitochondria. In fact, it has been observed that mitochondrial localization of EGFR and EGFRvIII appears to be important in modulating several cellular events, cell drug resistance, autophagy, and apoptosis [41], as well as for stimulating mitochondrial oxidative metabolism under low-glucose conditions, to sustain the proliferation of glioma cells [13]. In our previous study, we showed that, after 48 h, iPA at 10 μM was able to cause necroptosis in U87MG and cells over-expressing EGFRwt and EGFRvIII through the activation of necroptosis markers and the induction of PUMA. PUMA over-production is generally associated with mitochondrial localization and apoptosis, but one group reported interactions between PUMA and the cytosolic domain of the activated EGFR and EGFRvIII proteins, resulting in its cytosolic sequestration in glioblastoma cells, reduced levels of apoptosis, and increased cell survival [14,15]. Such evidence has led us to hypothesize a fascinating new role for iPA within glioblastoma cells as an inhibitor of EGFR signaling. 

In the current study, we have showed the ability of iPA to lower mitochondrial activity by preventing the translocation of EGFR/EGFRvIII on the mitochondria through the inhibition of Y845 phosphorylation. This event allows PUMA to interact with the organelle, thus triggering the necropoptotic or apoptotic process. Our data show that the growth suppression induced by iPA at 10 μM after 48 h is linked to metabolic alterations. Indeed, the results from OCR analysis indicated a strong reduction in cellular basal respiration, ATP production, and maximal respiratory capacity in GBM cells treated with iPA 10 μM for 18 h and 24 h as compared to untreated samples. The analysis by western blot assay of the total lysates of the GBM cells has highlighted that the iPA treatment did not change the levels of EGFR protein, but it induced PUMA protein levels. In addition, data from mitochondrial fractions revealed high levels of EGFRwt/EGFRvIII and absence of PUMA in untreated cells, while the iPA treatment reversed the mitochondrial localization of the proteins, suggesting the inhibition of the EGFR translocation on mitochondria and thus allowing the interaction of PUMA with the organelle. Such evidence was confirmed by immunofluorescence experiments on U87MG cells, in which EGFR signal co-localized with TOM20 signal in control samples, while in iPA-treated ones, the PUMA signal overlapped with TOM20. Indeed, the interaction of PUMA on mitochondria is linked with apoptotic or necroptotic events in several cancer cell lines [16,17]. 

Recent observations have demonstrated that EGFR/EGFRvIII was constitutively activated in the majority of GMB cases undergoing phosphorylation operated by several kinases, including Src kinase. This kinase is responsible of Y845 phosphorylation, which is important for the EGFR/EGFRvIII translocation to mitochondria, contributing to their integrity as much as to the survival of cancer cells [34]. We observed that the iPA treatment completely inhibited the phosphorylation of Y845 after 18 h, when the mitochondrial activity was reduced. Subsequently, to confirm whether EGFR translocation on mitochondria was dependent on the phosphorylation of Y845, the U87MG, U87MGEGFRwt, and U87MGEGFRvIII cells were transiently transfected with plasmids encoding a mutant receptor bearing a phenylalanine substitution at Y845 (EGFRY845F). Mitochondrial fraction obtained from GBM cells transfected with the Y845F variant of EGFR/EGFRvIII showed no detectable levels of EGFR protein, while PUMA protein was clearly increased (Figure 4B). Immunofluorescence experiments supported the inhibition of EGFR translocation in U87MG transfected with EGFRY845F, suggesting that mitochondrial localization of EGFR was dependent upon Y845 phosphorylation. Interestingly, the transient transfection of Y845F variant of EGFR/EGFRvIII in GBM cells determined a reduction in basal and maximal mitochondrial respiratory capacities, as revealed by OCR analysis. Lastly, one group described the Src-dependent phosphorylation of Y845 as important for the survival and proliferation of glioma cells through its involvement in the stimulation of mitochondrial oxidative metabolism [13]. 

Our findings demonstrate, for the first time, that the phosphorylation of this tyrosine residue of EGFR by Src Kinase is important for the mitochondrial metabolism; in fact, its inhibition by iPA treatment causes a reduction of the mitochondrial respiration. To better characterize the iPA effect on mitochondrial activity, we used an additional GBM cell line, U251MG, which is characterized by a mutated p 53 and is unable to activate the transcription of target genes such as PUMA. The analysis of western blot of total cell lysates did not reveal detectable level of PUMA, while EGFR protein levels did not change upon iPA treatment. The OCR experiments performed on U251MG treated with 10 μM iPA for 24 h showed no significant changes in the mitochondrial activity in comparison with untreated cells. Overexpression of PUMA and EGFR by transfection assay in these cells revealed a significant reduction of the basal and maximal mitochondrial respiratory capacities upon iPA treatment in comparison with those that were co-transfected with the control vector, suggesting the suppressive role of PUMA of the mitochondrial activity. To further investigate this event, we evaluated the effects of PUMA knockdown on the mitochondrial respiration in U87MG cells treated with or without iPA. Western blot of total lysates revealed that in U87MG transfected with siRNA-PUMA, the protein was not detectable after 24 h (Figure 5E) compared to the iPA-treated samples and the control transfected with scrambled siRNA. The OCR analysis of U87MG cells transfected with siRNA-PUMA and treated or untreated with iPA did not show significant variations in mitochondrial activity compared with control samples. In contrast, the iPA treatment of cells transfected with siRNA scramble totally reduced the mitochondrial activity. 

Lastly, to further demonstrate the protective role of EGFR and suppressive role of PUMA on mitochondrial activity, we performed over-expression of the Y845F variant of EGFR and PUMA in U251MG cells by transfection assay and then analyzed mitochondrial respiration. We observed that the oxygen consumption rate (OCR), an indicator of OXPHOS, was significantly inhibited in U251MG transfected cells. In support of these results, a recent and interesting study described the suppression of oxidative phosphorylation due to PUMA induction in hepatocarcinoma cancer cells [35]. Furthermore, recent evidence has shown that PUMA can promote changes in mitochondrial morphology, playing an important role in apoptosis or necroptosis [36]. Indeed, in one study, mitochondria dynamics were related to apoptosis, and the integrity of the mitochondrial membrane was destroyed when apoptosis was induced, resulting in lack of fusion or excessive fission of mitochondria and leading to mitochondrial fragmentation [42]. As important members of mitochondrial fusion, Mfn1 and fission protein family Fis1 are involved in the regulation process of cells apoptosis [37], so we investigated whether iPA treatment could modulate the expression of these proteins. Data obtained by western blot showed a significant increase in the fission protein levels of Fis1 and no change in the fusion protein levels Mfn1. Immunofluorescence experiments revealed changes in mitochondrial morphology, being fragmented in U87MG treated with iPA in comparison with the control sample. These interesting findings require further investigation to clarify the molecular mechanism related to the modulation of the proteins regulating the mitochondrial morphology. 

To prove that altered energy metabolism is a hallmark of GBM [43] and that it could be a potential therapeutic target, we studied the iPA effect on cell viability and mitochondria metabolism of GBM cells derived from affected patients; iPA treatment inhibited cell viability at 48 h, inducing cell death by apoptosis of all primary GBM cell models used. Afterwards, we studied the mitochondria activity by OCR analysis. We ascertained that GBM cell models exhibited elevated bioenergetic demands compared with human astrocyte cell lines and that iPA treatment at 10 μM for 24 h disrupted GBM cell energy metabolism. The molecular mechanism appears to be similar to that found in GBM cell lines. In fact, we observed induction of PUMA and inhibition of Y845 phosphorylation of EGFR. In particular, transient transfection of a Y845F variant of EGFR into a GBM cell model, GBM4, caused a reduction of mitochondrial respiration. 

Nonetheless, there are some limitations to our study. For example, iPA is an experimental/preclinical compound that requires chemical and pharmacological optimization before it can be tested in clinical applications, and the inhibitory effect of iPA on EGFR signaling remains to be further investigated. Overall, this study highlighted how this experimental small molecule targets molecular vulnerability linked to energy metabolism in GBM cells by inhibiting the Y845 phosphorylation of EGFR and the following translocation on mitochondria, unlike other EGFR signaling inhibitors that increase mitochondrial translocation after treatments, such as staurosporine (ST) and Iressa (I), limiting their clinical efficacy [14]. These findings remark the potential for using iPA for therapeutic developments.

## 5. Conclusions

This novel mechanism of action for targeting tumor cells, together with the other pleiotropic effects of iPA on cancer growth such as angiogenesis inhibition [19] and immune anti-tumor responses stimulation mediated by NK-T [20], radio sensitization effects [21], and the lack of cytotoxicity on noncancerous cells, open the way to new drugs targeting RTKs; iPA alone or in combination with other anti-EGFR therapies can provide a rationale for a more favorable targeted therapy in GBM treatment.

## Figures and Tables

**Figure 1 cancers-14-06044-f001:**
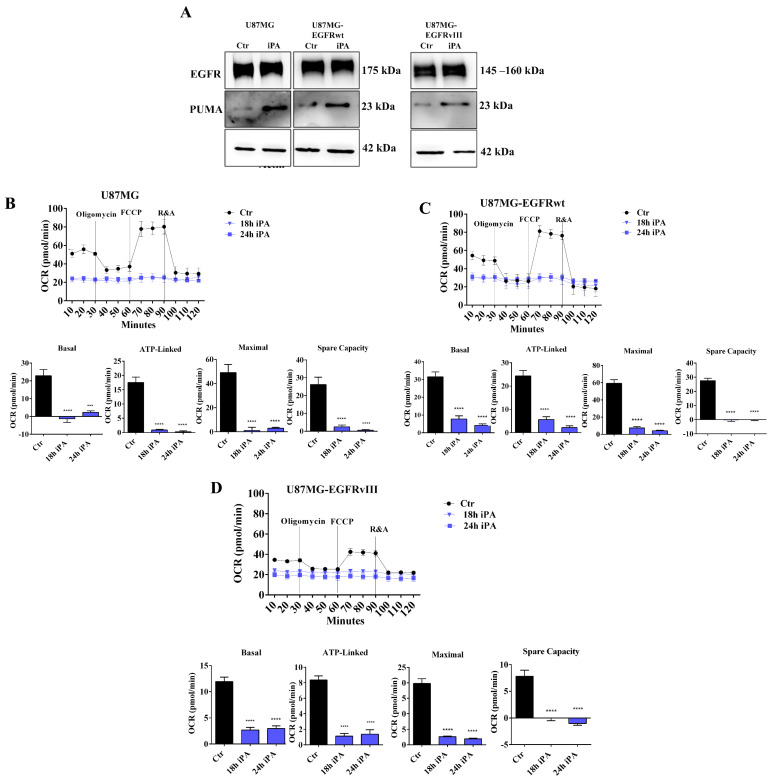
iPA impaired mitochondrial metabolism in GBM stabilized cell lines. (**A**) Representative immunoblotting images of EGFR and PUMA protein levels in U87MG, U87MG–EGFRwt, and U87MG–EGFRvIII cells after iPA treatment. Kinetic profile of oxygen consumption rate (OCR) of U87MG (**B**); U87MG–EGFRwt (**C**); U87MG–EGFRvIII (**D**) cells, treated or untreated with iPA for 18 or 24 h. Parameters of the mitochondrial respiration pathway calculated from OCR profile: basal respiration (calculated as the difference between OCR before the first injection and OCR after Ant + Rot injection), maximal respiration (calculated as the difference between OCR after FCCP injection and OCR after Ant + Rot injection), ATP production (calculated as the difference between OCR before oligomycin injection and OCR after oligomycin injection), spare capacity (calculated as the difference between maximal respiration and basal respiration). Data are from 3 independent experiments that were conducted in technical replicate. Data are expressed as mean ± SEM. Data were analyzed for statistical significance using the 2-tailed Student t-test or ANOVA following by Bonferroni correction for multiple comparisons. Results are representative of 3 experiments performed in duplicate and are expressed as mean ± SD (ANOVA *** *p* < 0.001, **** *p* < 0.0001).

**Figure 2 cancers-14-06044-f002:**
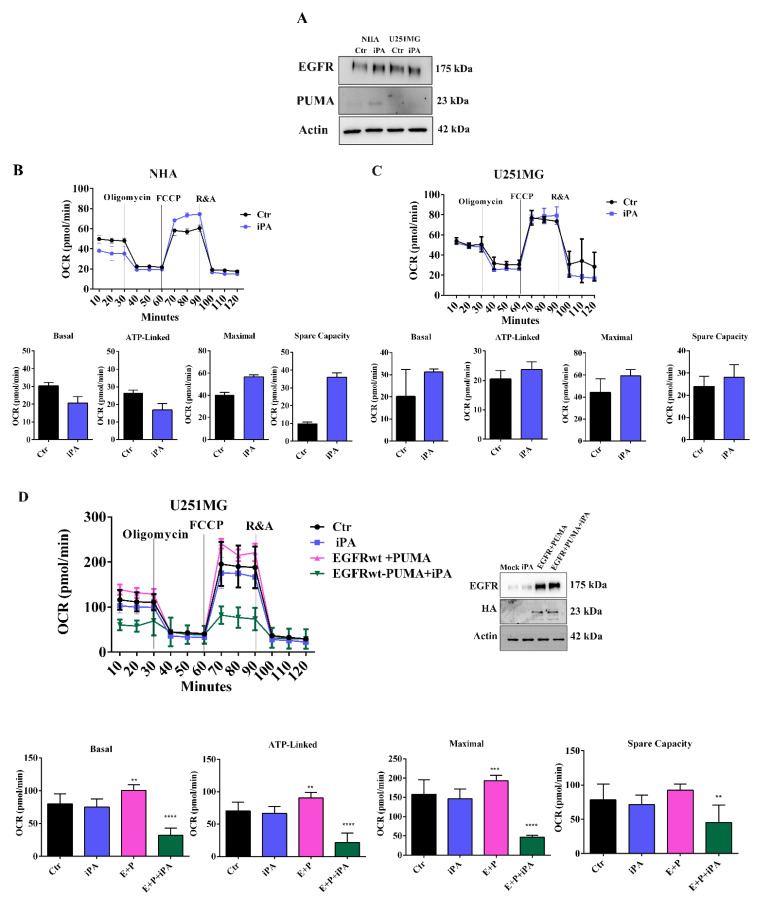
iPA impairs mitochondrial function and glycolysis in NHA cells and U251MG cells. (**A**) Immunoblot showing the effect of iPA treatment on EGFR and PUMA protein levels in NHA cells. Immunoblot showing the effect of iPA treatment on EGFR protein levels in U251MG cells. (**B**) OCR rates in NHA cells show that the mitochondrial activity was not affected after iPA treatment. (**C**) OCR rates in U251MG were reduced by iPA treatment. (**D**) **Left**: OCR in U251MG cells treated or not with iPA 10 µM and the same cells transfected with PUMA and EGFR, treated or untreated with iPA 10 µM. **Right**: western blot analysis showing protein levels of EGFR and PUMA after transfection in U251MG cells transfected with EGFRwt + PUMA, treated or untreated with iPA 10 µM for 24 h. Data are from three independent experiments in technical replicate. Data are expressed as mean ± SEM. Data were analyzed for statistical significance using the 2-tailed Student *t*-test or ANOVA following by Bonferroni correction for multiple comparisons. Results are representative of 3 experiments performed in duplicate and are expressed as mean ± SD (ANOVA ** *p* < 0.01; *** *p* < 0.001, **** *p* < 0.0001).

**Figure 3 cancers-14-06044-f003:**
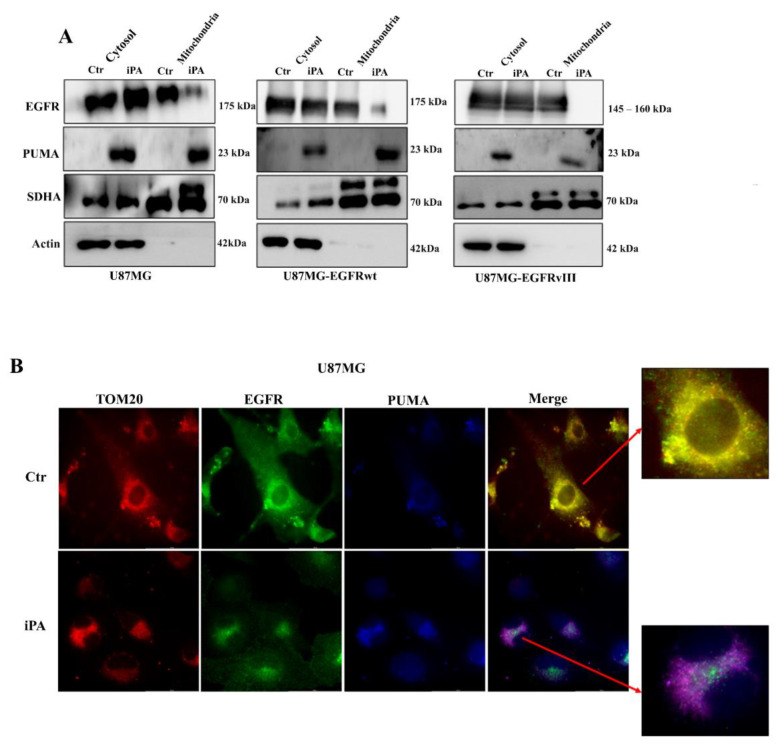
iPA inhibits the EGFR/EGFRvIII translocation on mitochondria in GBM cells. (**A**) Western blot showing the distribution of EGFR and PUMA levels in mitochondrial fraction and cytosolic fraction, showing the effects of iPA 10 µM on EGFR and PUMA subcellular localization. (**B**) Immunofluorescence analysis of EGFR–PUMA localization in U87MG cells that had been treated and untreated with iPA 10 µM. U87MG cells were stained with EGFR antibody (green) and PUMA antibody (blue), while the mitochondria were stained using the primary antibody TOM20 (red). The yellow color of merged imaged of Ctr show the co-localization of EGFR and TOM 20 (overlapping od red+ green signals; red arrow indicating the magnification marge image). The merged images of U87MG cells treated with iPA 10 µM produced a violet color, suggesting a co-localization of TOM20 and PUMA (overlapping of red + blue signals; red arrow indicating the magnification marge image). Data are from three independent experiments that were conducted in technical replicate. Data are expressed as mean ± SEM. Data were analyzed for statistical significance using the 2-tailed Student *t*-test or ANOVA, followed by Bonferroni correction for multiple comparisons. Results are representative of 3 experiments performed in duplicate and are expressed as mean ± SD.

**Figure 4 cancers-14-06044-f004:**
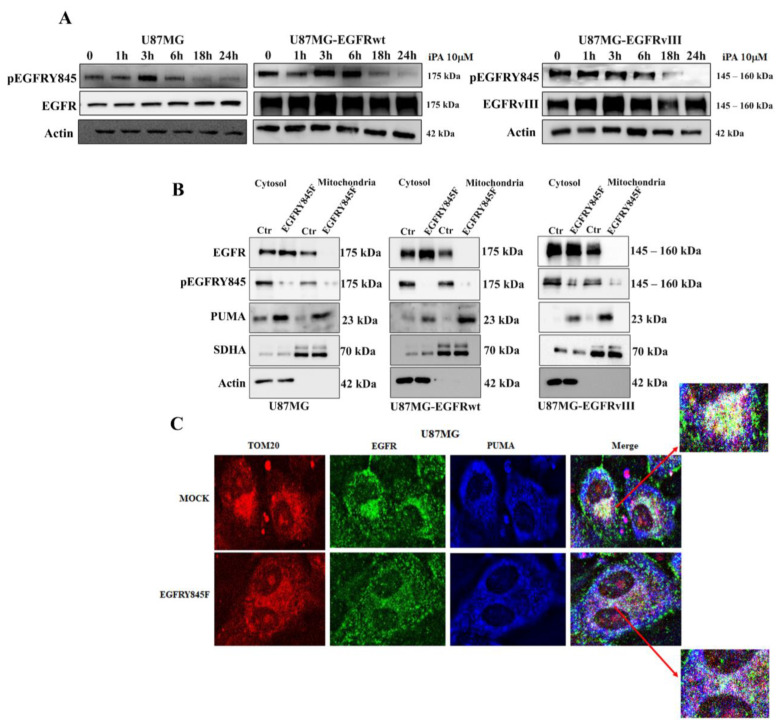
iPA inhibits the Y845 phosphorylation of EGFR and EGFRvIII in GBM cells. (**A**) Western blot showing the effects of iPA 10 µM on the levels of pEGFR (Y845) and EGFR in U87MG, U87MG–EGFRwt, and U87MG–EGFRvIII cells at different time points. The data show a reduction of the phosphorylation starting at 18h of treatment. (**B**) Western blot analysis on mitochondria and cytosolic fractions transfected with EGFRY845F and treated with iPA 10 µM. (**C**) Immunofluorescence staining showing the different cellular localization of EGFR and PUMA in U87MG cells control (MOCK) and transfected with EGFRY845F.The EGFR signal (green) overlapped the TOM20 signal (red), producing a light-yellow color (red arrow indicating the magnification marge image); in U87MG cells transfected with EGFRY845F plasmid, we observed the most widespread EGFR signal (green), and the EGFR signal did not overlap the TOM20 signal (red) (merge image). Conversely, the PUMA signal (blue) was increased and overlapped the TOM20 signal (red), producing a violet color in the cytosol (red arrow indicating the magnification marge image). Data are from three independent experiments in technical replicate. Data are expressed as mean ± SEM. Data were analyzed for statistical significance using the 2-tailed Student *t*-test or ANOVA following by Bonferroni correction for multiple comparisons. Results are representative of 3 experiments performed in duplicate and expressed as mean ± SD.

**Figure 5 cancers-14-06044-f005:**
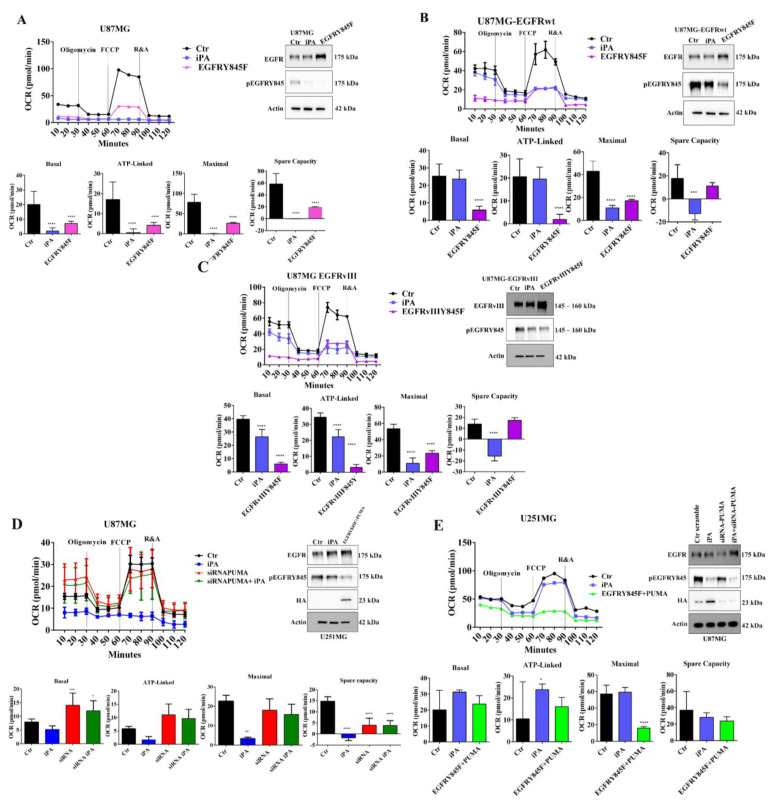
The Y845 phosphorylation of EGFR is important for mitochondrial activity. (**A**–**C**) **Left**: OCR schemes of U87MG, U87MG–EGFRwt, and U87MG-EGFRvIII treated with or without 10 µM iPA and the same cells transfected with EGFRY845F. **Right**: western blot analysis showing protein levels of EGFR and pEGFRY845 in U87MG, U87MG-EGFRwt, and U87MG-EGFRvIII cells transfected with EGFRY845F. (**D**) **Left**: OCR in U251MG treated or untreated with 10 µM iPA and the same cells transfected with EGFR Y845F + PUMA compared to the cells transfected with control vector. **Right**: western blot analysis showing protein levels of EGFR in U251MG cell lines transfected with EGFRY845F + PUMA compared to the cells transfected with control vector. (**E**) **Left**: OCR of U87MG treated or not with 10 µM iPA and the same cells transfected with PUMA siRNA. **Right**: western blot analysis showing protein levels of EGFR, pEGFRY845, and PUMA in U87MG lines transfected with PUMA siRNA. Data are from three independent experiments in technical replicate. Data are expressed as mean ± SEM. Data were analyzed for statistical significance using the 2-tailed Student *t*-test or ANOVA followed by Bonferroni correction for multiple comparisons. Results are representative of 3 experiments performed in duplicate and are expressed as mean ± SD (ANOVA * *p* < 0.05; ** *p* < 0.01; *** *p* < 0.001, **** *p* < 0.0001).

**Figure 6 cancers-14-06044-f006:**
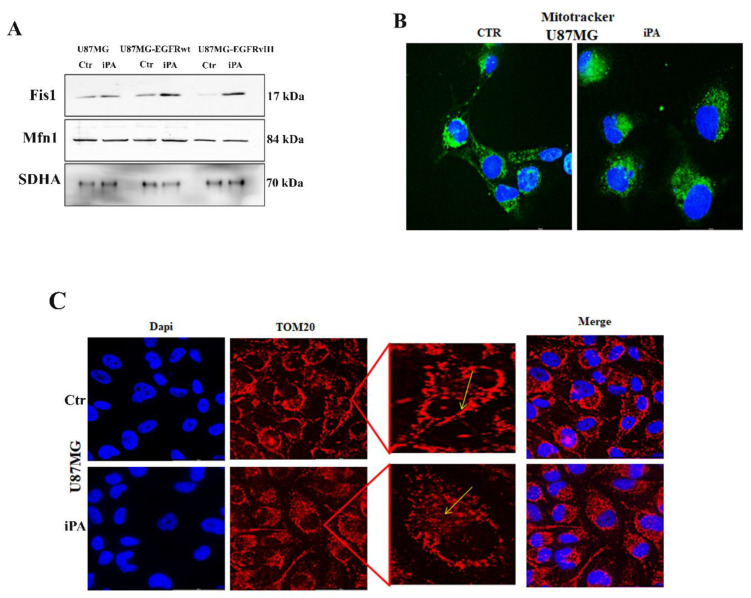
iPA interferes with mitochondrial dynamics. (**A**) Western blot of Fis1 and Mfn1 in U87MG, U87MG-EGFRwt, U87MG-EGFRvIII. (**B**) Mitochondria staining in U87MG cell line using DAPI to stain nuclei and MitoTracker (Green) to stain mitochondria. MitoTracker staining shows that mitochondrial shape was changed by iPA 10 µM treatment. (**C**) Immunofluorescence analysis of mitochondria using primary antibody TOM20 (red), while the nuclei were stained with DAPI. The images show that U87MG cells treated with iPA 10 µM (lower panel) contained more round or fragmented mitochondria, while untreated cells (upper panel) contained more fused or elongated mitochondria (mitochondrial morphological changes are indicated by the yellow arrow). Data were analyzed for statistical significance using the 2-tailed Student *t*-test or ANOVA following by Bonferroni correction for multiple comparisons. Results are representative of 3 experiments performed in duplicate and are expressed as mean ± SD.

**Figure 7 cancers-14-06044-f007:**
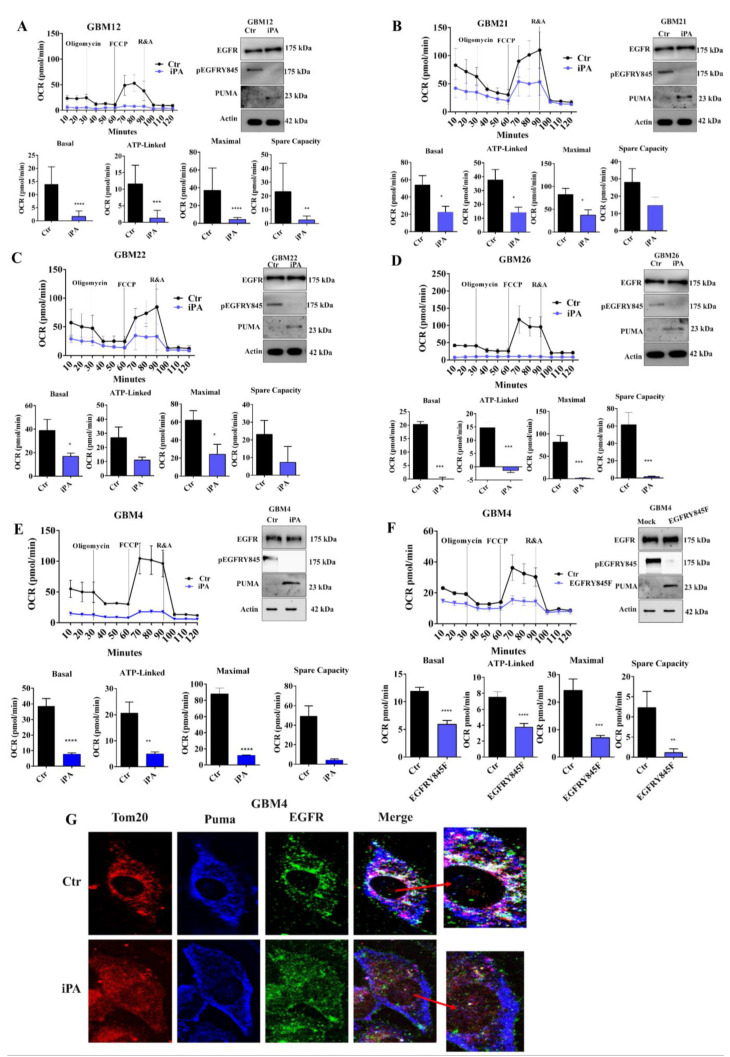
iPa effects on mitochondrial metabolism on primary glioblastoma cells. Left panels of (**A**–**E**) are the representative OCR showing the inhibitory effect of 10 µM iPA on mitochondria functionality in six primary glioblastoma cell lines (GBM12 to GBM4). On the right of each plot is shown the immunoblotting analysis for the EGFRY845 and PUMA levels. 10 µM iPA reduced the EGFRY845 levels, while PUMA levels were increased. (**F**) **Left**: OCR of GBM4 transfected with EGFRY845F or not. **Right**: western blot analysis showing EGFR and pEGFRY845 protein levels after 10 µM iPA on GBM4 cells that were or were not transfected with EGFRY845F. (**G**) Immunofluorescence staining of GBM4 cells showing the different localization of EGFR and PUMA after 10 µM iPA treatment (lower panel). GBM4 cells showed an increased blue signal (PUMA), while in merged images, TOM20 co-localized with PUMA, producing a violet color due to the reduction of the EGFR green signal as compared to Ctr (red arrow indicating the magnification marge image). Data were analyzed for statistical significance using the 2-tailed Student *t*-test or ANOVA, followed by Bonferroni correction for multiple comparisons. Results are representative of 3 experiments performed in duplicate and are expressed as mean ± SD (ANOVA * *p* < 0.05; ** *p* < 0.01; *** *p* < 0.001,**** *p* < 0.0001).

**Table 1 cancers-14-06044-t001:** Patients and primary cell line characterization: Genetic and epigenetic profile of tissues obtained by surgery compared with derived primary cells. MGMT methylation status, IDH1 and IDH2 mutation status, and evaluation of co-deletion 1p–19q were extrapolated from the EPIC (850 K) Methylome Array. The data were confirmed by independent techniques, used as gold standard. Epigenetic sub-class was obtained comparing sample methylome profile with samples present in TGCA database bank. Epigenetic subclasses indicate mesenchymal GMB as a gliosarcoma; RTKI as a proneural tumor; RTKII as a classical or neural tumor. Integrated genomic analysis identifies clinically relevant subtypes of glioblastoma characterized by abnormalities in PDGFRA, IDH1, EGFR, and NF1. Cancer Cell 17, 98–110 (2010).

ID	Patient		Type	MGMT Methylation		IDH1/IDH2 Mutation		Co-del-1p-19q	Grading	EGFR Amplification	Epigenetic Subclass
	** *Age* **	** *Gender* **		** *Biopsy* **	** *Primary Cell-Line* **	** *Biopsy* **	** *Primary Cell-Line* **	** *Biopsy* **	** *Biopsy* **	** *Biopsy* **	** *Biopsy* **
**GBM4**	64	M	Primitive	Un-meth	Meth	Wt	Wt	No	GBM (WHO IV)	no	Mesenchymal
**GBM12**	60	F	Primitive	Meth	Un-Meth	Wt	Wt	No	GBM (WHO IV)	yes	RTKII
**GBM21**	66	M	Primitive	Meth	Un-Meth	Wt	Wt	No	GBM (WHO IV)	yes	RTKI
**GBM22**	70	M	Primitive	Un-Meth	Un-Meth	Wt	Wt	No	GBM (WHO IV)	weak	RTKI
**GBM26**	50	M	Primitive	Meth	Meth	Wt	Wt	No	GBM (WHO IV)	weak	Mesenchymal

## Data Availability

The data presented in this study are available on request from the corresponding author.

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
