# Peer review of "N6-Isopentenyladenosine Impairs Mitochondrial Metabolism through Inhibition of EGFR Translocation on Mitochondria in Glioblastoma Cells"

_cancers, 2022, doi:10.3390/cancers14246044_

Round 1

Reviewer 1 Report

Review Report is attached.

Author Response

Dear Reviewer 1,

I am re-submitting the manuscript entitled: “N6-isopentenyladenosine impairs mitochondrial metabolism through inhibition of EGFR translocation on mitochondria in glioblastoma cells” by Pagano et al., for consideration as a Full paper in Cancers.

We thank the reviewer for his careful manuscript reading and suggestion.

The manuscript has been revised according to the editor and reviewers' comments and uploaded.

Changes should be easily visible to the editors and reviewers since they were clearly highlighted in yellow. Original reviewer comments in boldface.

Here I am providing point-by-point the details of the revisions in the manuscript and the responses to the reviewers' comments:

Response to Reviewer 1

In the final western blot panels shown in the manuscript figures we added a single actin as a representative of the loading control. In the Raw Data file, we presented the loading control for each target analyzed by western blot. We have also added the fractionation control, when necessary. Each western blot was loaded on the same SDS-PAGE gel as per suggestion.

Reviewer: Major Concern 1. The localization experiments performed by the authors in the manuscript hold a major concern. The protein blots for different fractions have been shown separately which makes It difficult to compare the purity of the fraction and thus make any further conclusions on the localization of different entities.

Authors’ response: Thank you for the suggestion. We loaded onto a single western blot the cytosolic and mitochondrial fractions for each cell line with the fractionation control SDHA and cytosolic control Actin, which is not found in the mitochondria.

Minor Concern 1. In Figures 1A, B, and C, the authors should load the samples from U87MG-WT & the over-expressing versions of the EGFR variants on the same SDS-PAGE gel so that an efficient conclusion on the over-expression of EGFR can be made accurately.

Authors’ response 1: Figures 1A, B and C became Figure 1A after modifications. We loaded the different samples on the same SDS-PAGE gel as it was suggested. In the Raw Data file, we added the western blot of EGFR of U87MG, U87MG-EGFRwt and U87MG-EGFRvIII as a single image, since we loaded and run the six samples together. In Figure 1A we separated the samples and added the better exposed lanes of U87MG to better show the signal coming from U87MG samples. This is due to the fact that the EGFR lanes of U87MG-EGFRwt and U87MG-EGFRvIII were particularly intense, hiding the U87MG signal. We exposed the U87MG lanes again in order to better analyse and see the signal.

Minor  Concern  2.  In  Fig  2A,  the  authors  have  not  provided  the  western  blot  image  for  the expression of PUMA in U251MG cells with and without iPA treatment. Moreover, it is important to load  the  protein  lysates  from  the  two  cell  lines  for  this  experiment  on  the  same  western  blot membrane to assess the level of expression of both PUMA & EGFR after the iPA treatment.

Authors’ response 2: We ran the experiment as per the reviewer’s suggestion and presented the data in the new Raw Data file. Furthermore, in the U251MG cell line, Puma is undetectable, reason why we previously decided to not include it in the final figure. Figure 2A and 2B merged into Figure 2A after modifications.

Minor Concern 3. In Figure 2E legend, the authors have mislabeled U251MG cells as U251. Please change.

Authors’ response 3: Thank you, this has been corrected. Figure 2E became Figure 2D after modifications.

Minor Concern 4. In Fig, 2E, the transfection of U251MG cells with EGFR and PUMA together did not  increase  the  mitochondrial  activity  beyond  the  transfection  control.  Do  the  authors  have  an explanation to this observation? It seems like U251MG cells are not dependent on PUMA & EGFR-WT for their mitochondrial activity. Although, after treatment of EGFR+PUMA transfected U251MG cells with the iPA, the mitochondrial activity drops. The authors should explain this phenomenon.

Authors’ response 4: Thank you for the question, we would like to clarify this point. We used U251MG as a cell model in which p53 is mutated in the DNA binding domain, which is the reason why PUMA, an important target gene of p53, is low expressed. We decided to use U251MG to demonstrate the protective role of EGFR on the tumor homeostasis. As we observed, iPA treatment inhibits EGFR translocation to the mitochondria, allowing PUMA to interact with the organelle thus triggering cell death pathways (apoptosis or necroptosis as reported in Pagano et al. doi: 10.1038/s41420-022-00974-x). In addition, the efficient mitochondrial activity of U251MG, as it can be seen by the OCR analysis, is due to the presence of EGFR and the basal reduced PUMA expression. With the experiment shown in figure 2E, we wanted to demonstrate how iPA, in the presence of both transfected EGFR and PUMA, is capable of reducing mitochondrial activity by activating the above-mentioned mechanism. We hypothesize that in the transfected samples the OCR does not vary due to the protective role of EGFR, demonstrating that it is indeed PUMA that can induce the reduction of mitochondrial activity triggering cell death. Figure 2E became Figure 2D after modifications.

Minor  Concern  5.  In  Figures  3A,  B,  and  C,  the  authors  should  have  loaded  the  samples  for mitochondria & cytosol on the same SDS-PAGE and blotted them on the same membrane to compare the expression of the respective markers, SDHA & Actin. This is a very important step in comparing the purity of the fractionation. The subsequent results in the figure cannot  be conclusive unless the fractionation step has been carefully addressed in the second chapter of the results section.

Authors’ response 5: Thank you for the suggestion, we ran the experiment as per request and presented the data in the new Raw Data file. Figure 3A and 3B became 3A; 3C became 3B.

Minor Concern 6.  In Figure 3C, EGFR, PUMA, and mitochondrial marker TOM20 can be seen colocalized together with or without iPA treatment and do not reflect the conclusions made in Figures 3A, and B. The authors need to clarify this chapter again before making strong conclusions.

Authors’ response 6: Thank you for identifying this area of potential ambiguity. We would like to clarify this point. Our microscopy analysis wanted to demonstrate a regional colocalization, which is presented as a yellow color, between the mitochondria, marked with TOM20 (red signal), and EGFR (green signal). After iPA treatment, PUMA (blue signal) increases and translocate on the mitochondria, producing a purple color (violet) due to its overlapping on the red signal produced by TOM20. At the same time, we observed a reduction in the EGFR translocation on the mitochondria, which translates to an overall reduction of the green signal from the region (mitochondria). We would like to highlight that the colocation of EGFR-PUMA is outside the scope of this paper, since it has already been demonstrated previously by Zhu et al (DOI: 10.1016/j.canlet.2010.01.028); our aim was to demonstrate an inhibition of EGFR Y845 phosphorylation, which is the principal mechanism that we demonstrated hinders the receptor translocation on the mitochondria, favoring the PUMA interaction with the organelle. Figure 3C became 3B.

Minor Concern 7.  The results specified in Fig 4B  could be very descriptive if the authors would have loaded the samples from the fractionation on the same SDS-PAGE and would have also stained the mitochondrial and cytosolic fraction for SDHA & ACTIN to analyze the purity of the fractions.

Authors’ response 7: Thank you for the suggestion, we ran the experiment as per request and presented the data in the new Raw Data file.

Minor Concern 8. In Figure 4C, EGFR antibody staining fails to show any over-expression of mutant EGFR. The TOM20 signal also gets dispersed after mutant EGFR expression which makes it difficult to conclude whether it’s the effect of less EGFR localization to the cytosol or the distribution of mitochondria itself.

Authors’ response 8: Thank you for the comment. We did not intend to use the immunofluorescence analysis as a quantitative method. With this method we wanted to show the different subcellular localization of EGFR rather than establish the amount of receptor, which we preferred to demonstrate through the use of the Western blot. In figure 4C, after the transfection of EGFRY845F we observed the same morphology alteration of the mitochondria as we observed in the cells treated with iPA depicted in Figure 6 (B; C).

Minor  Concern  9.  In  Figure  5,  the  authors  have  used  the  pEGFRY845  antibody  to  detect  the phosphorylation status of the EGFR residue. However, even in the presence of endogenous EGFR, the authors  observe  the  complete  reduction  of  pEGFRY845  levels  in  U87MG  cells  transfected  with EGFRY845F. Why is it so?

Authors’ response 9: Thank you for highlighting this problem. The efficiency of the transfection was equal to 90%, thus minimizing the effect of the endogenous receptor.

Minor Concern 10.  In  Figure 5A,  the  authors observe  very  less  mitochondrial  activity  in U87MG cells with EGFRY845 transfection even when the cells express  the endogenous EGFR.  The authors should explain this phenomenon.

Authors’ response 10: Thank you, we would like to further explain this phenomenon. We performed a titration of the amount of transfected DNA of the mutant receptor (EGFRY845F); we chose the DNA quantity that had an efficiency equal to 90%, thus minimizing the effect of the endogenous receptor.

Minor Concern 11.  The samples in  Figure 6A  need  to be  loaded on the sample SDS-PAGE and stained together for better comparison of fractionation purity.

Authors’ response 11: Thank you for the suggestion, we ran the experiment as per request and presented the data in the new Raw Data file.

Minor Concern 12. PUMA protein gel image is missing for GBM4 treatment in Fig 7F.

Authors’ response 12: Thank you, this has been corrected.

Reviewer 2 Report

Pagano et al, proposing N6-isopentenyladenosine (iPA) as a novel therapeutic drug for glioblastoma multiforme. Authors observed treatment of iPA on GBM and other cancer cells inhibits mitochondrial metabolism and affect EGFR translocation into mitochondria. Further, they also found that it promote PUMA mediated cell death. Though the study seems to be interesting, it has many concerns regarding the original western blots provided and data representation. 

1.  Figure 1 A, B and C, all the samples should be run on the same gel parallelly and should be probed for different antibodies. ideally the extent of overexpression of EGFR WT compared to basal level should be determined. the original blots provided for this figure looks so unconvincing. some exposures have 8 lanes and some has four lanes. for PUMA blots, one set was picked up from last two lanes and other sets picked up from the middle two lanes. 

2. Figure 1 F, expression of EGFRvIII decreased basal OCR levels compared to untransfected and EGFR WT over expressed cells. can authors explain why? 

3. Figure 2 E, same comment as first point, EGFR and actin was taken from the middle lanes whereas HA-tagged PUMA was taken from left last four lanes. blots should be shown from the same membrane, if it was run on duplicates actin should be provided for the duplicate membrane too. 

4. Historically, if you are studying mitochondrial translocation of a protein or comparing cytosol proteins with mitochondrial fractions, the fractions should be run parallelly. SDHA runs at different size for U87MG compared to other two samples. why? 

5. Figure 3 C, authors should calculate colocalization co-efficient by doing quantitation for overlapping foci both Tom20/EGFR and PUMA/EGFR. localization can be zoomed in and shown as a separate box. Ctrl Tom20 staining has more background than iPA treated cells. were they taken with same laser power and exposure?

with the image provided, its hard to comment on specific co-localisation

can authors do immunoprecipitation and show more EGFR association with PUMA in presence and absence of iPA?

6.  Figure 4 B and C, same comment as 4 and 5

7. Figure 7, same problem as before. blots picked up from different membranes 

Author Response

Dear Reviewer 2,

I am re-submitting the manuscript entitled: “N6-isopentenyladenosine impairs mitochondrial metabolism through inhibition of EGFR translocation on mitochondria in glioblastoma cells” by Pagano et al., for consideration as a Full paper in Cancers.

We thank the reviewer for his careful manuscript reading and suggestions. The manuscript has been revised according to the editor and reviewers' comments and uploaded.

Changes should be easily visible to the editors and reviewers since they were clearly highlighted in yellow. Original reviewer comments in boldface.

Here I am providing point-by-point the details of the revisions in the manuscript and the responses to the reviewers' comments:

Response to Reviewer 2

In the final western blot panels shown in the manuscript figures we added a single actin as a representative of the loading control. In the Raw Data file, we presented the loading control for each target analyzed by western blot. We have also added the fractionation control, when necessary. Each western blot was loaded on the same SDS-PAGE gel as per suggestion.

Pagano et al, proposing N6-isopentenyladenosine (iPA) as a novel therapeutic drug for glioblastoma multiforme. Authors observed treatment of iPA on GBM and other cancer cells inhibits mitochondrial metabolism and affect EGFR translocation into mitochondria. Further, they also found that it promote PUMA mediated cell death. Though the study seems to be interesting, it has many concerns regarding the original western blots provided and data representation.

Reviewer: 1.  Figure 1 A, B and C, all the samples should be run on the same gel parallelly and should be probed for different antibodies. ideally the extent of overexpression of EGFR WT compared to basal level should be determined. the original blots provided for this figure looks so unconvincing. some exposures have 8 lanes and some has four lanes. for PUMA blots, one set was picked up from last two lanes and other sets picked up from the middle two lanes.

Authors’ response: Figures 1A, B and C became Figure 1A after modifications. Thank you for the suggestion. We ran the experiment as per the reviewer’s suggestion and presented the data in the new Raw Data file.

Reviewer: 2. Figure 1 F, expression of EGFRvIII decreased basal OCR levels compared to untransfected and EGFR WT over expressed cells. can authors explain why?

Authors’ response: Figure 1F became 1D after modifications. We thank you for this question. We cannot comment yet on this type of data as it is one of the first data obtained in the literature. It seems premature to comment on it as we are still working on it.

Reviewer: 3. Figure 2 E, same comment as first point, EGFR and actin was taken from the middle lanes whereas HA-tagged PUMA was taken from left last four lanes. blots should be shown from the same membrane, if it was run on duplicates actin should be provided for the duplicate membrane too.

Authors’ response: Thank you for the suggestion. We ran the experiment as per the reviewer’s suggestion and presented the data in the new Raw Data file. Figure 2E became 2D after modifications.

Reviewer: 4. Historically, if you are studying mitochondrial translocation of a protein or comparing cytosol proteins with mitochondrial fractions, the fractions should be run parallelly. SDHA runs at different size for U87MG compared to other two samples. why?

Authors’ response: We loaded the different samples on the same SDS-PAGE gel as it was suggested.

Reviewer: 5. Figure 3 C, authors should calculate colocalization co-efficient by doing quantitation for overlapping foci both Tom20/EGFR and PUMA/EGFR. localization can be zoomed in and shown as a separate box. Ctrl Tom20 staining has more background than iPA treated cells. were they taken with same laser power and exposure?with the image provided, its hard to comment on specific co-localization can authors do immunoprecipitation and show more EGFR association with PUMA in presence and absence of iPA?

Authors’ response: Figure 3C became 3B after modifications. We thank the reviewer for the valuable suggestion. We calculated a colocalization coefficient for TOM20 and EGFR to express the correlation intensity between the two-color channels. The Pearson’s coefficient (https://doi.org/10.1016/j.ymeth.2017.01.005) have been calculated using ImageJ software for images 3B, 4C and 7G: Pearson’s coefficient resulted in ≥ 1 Ctr vs iPA showing an increased colocalization in Ctr samples. As for the CTR TOM20 image in Figure 3B, we did not change the laser power and exposure settings; the image was correctly replaced. We added the zoomed in quadrant in the microscopy images showing the colocalization. Furthermore, we would like to clarify that the aim of our manuscript is to demonstrate the colocalization of EGFR/mitochondria. This is the reason why we decided not to study the interaction between EGFR/PUMA, especially through a co-immunoprecipitation analysis. The EGFR-PUMA interaction has already been demonstrated in a different manuscript written by Zhu et al (DOI: 10.1016/j.canlet.2010.01.028).

Reviewer: 6.  Figure 4 B and C, same comment as 4 and 5        

Authors’ response: We loaded the different samples on the same SDS-PAGE gel as it was suggested. We have included the colocalization coefficients in the previous response.

Reviewer: 7. Figure 7, same problem as before. blots picked up from different membranes

Authors’ response: Thank you for the suggestion. We ran the experiment as per the reviewer’s suggestion and presented the data in the new Raw Data file.

Round 2

Reviewer 2 Report

Authors addressed the concerns efficiently which improved the manuscript for publication. there are some corrections needs to done before publication. 

1.  SDHA is an inner mitochondrial protein. why it is enriched in cytosol fraction? was it flipped? 

Figure 3A right panel has SDHA in all fractions. 

Author Response

Dear Reviewer 2,

We thank you again for reviewing our manuscript and the revisions we have made thus far. We'll get back to you with the second round of suggestions.

Unfortunately, in the making of our new raw data file, due to the large number of new experiments done this month, we did not properly control and flipped the SDHA blots for Figure 3A and Figure 4B. We have corrected the error and sent a new raw data file and correctly assembled figures.

For mitochondrial fractionation analysis, we performed a purification to ensure that, after iPA treatment, EGFR was no longer detectable in mitochondria. Although the various purification steps were performed successfully, SDHA was still found in the cytosol fraction. However, we considered the obtained data reliable and considered the fractionation analysis reliable since β-actin was not detectable in the mitochondrial fraction, which we wanted to be particularly purified for the purposes of our manuscript and this is also especially true for Figure 3A.

Considering how important your suggestions have been to improve our work, if you feel that these data could confuse the reader about our results, our solution could be to show only the mitochondrial fraction to better explain our results, leaving the fractionation of the cytosol in the raw data as proof of the accuracy of our data.
